# Opening research sites in multicentre clinical trials within the UK: a detailed analysis of delays

Anna Kearney,[1] Andrew McKay,[1] Helen Hickey,[1] Silviya Balabanova,[1] Anthony G Marson,[2] Carrol Gamble,[3] Paula Williamson[3]

▶ Prepublication history and additional material is available. To view please visit the journal (http://dx.doi.org/10.1136/bmjopen-2014-005874).

[1]Clinical Trials Research Centre, University of Liverpool c/o Institute of Child Health, Alder Hey Children's NHS Foundation Trust, Liverpool, UK.
[2]Department of Molecular and Clinical Pharmacology, University of Liverpool, Liverpool, UK
[3]Department of Biostatistics, University of Liverpool, Liverpool, UK

**Correspondence to**
A Kearney;
a.kearney@liv.ac.uk

## ABSTRACT

**Objective:** To investigate the length of time taken to open UK research sites in multicentre clinical trials and to identify reasons for any delays.

**Design:** A case study, recording key milestone dates from the time a site receives ethical approval through to opening to recruitment. Delay reasons were prospectively logged by trial staff at a minimum of fortnightly periods using a coding system.

**Setting:** SANADII, a phase IV pragmatic trial, managed by the Clinical Trials Research Centre, Liverpool. The trial seeks to work with over 100 National Health Service (NHS) sites to meet its recruitment target of 1510 patients.

**Outcomes and analysis:** The primary outcome was time from Multicentre Regional Ethics Committee (MREC) approval to site opening using survival analysis. Where sites took over a specified time to reach milestones (>3 months from MREC to Site Specific Information (SSI) submission, >30 days from SSI validation to local R&D approval, or >30 days from local Research and Development (R&D) approval to opening to recruitment), the longest continuous delay during that milestone was identified.

**Results:** The median opening time for participating sites was 9.7 months (IQR 6.2 to Not Reached). SSI submission took 7 months (IQR 4.1–12.3) from ethics approval, R&D approval took 16 days (IQR 5.0–32.0) from SSI validation and site opening took 15 days (IQR 8.5–40.0) following R&D approval. The longest delays before SSI submission resulted from negotiating excess treatment costs, finalising logistics, collecting CVs and ongoing participation discussions.

**Conclusions:** While recently imposed targets are reducing the time taken for R&D departments to approve valid applications, the time taken to open UK research sites remains excessive and must be reduced. At present significant public funds are being used inefficiently in order to navigate NHS systems, challenging the resolve of trial teams and the competitiveness of the UK.

## Strengths and limitations of this study

- The prospective evaluation of time taken to open UK research sites using clearly defined milestone dates and coded delay reasons allows a detailed analysis of the current barriers to initiating trials in the UK.
- We identified that current targets are reducing the time taken to approve local National Health Service (NHS) applications but that the overall time remains excessive.
- The identification of delay reasons can strongly contribute to discussion around the Health Research Authorities development of a single approval system.
- The analysis of delay reasons is limited to the perspective of the Clinical Trial Research Centre based on information available from day to day trial management.

research sites in multicentre clinical trials, which increases the cost to funders and makes the UK a less attractive location to conduct research.[1][2] The current governance process for UK clinical trials requires approval from an ethics committee (and other regulatory authorities depending on the intervention), as well as obtaining separate local Research and Development (R&D) approval from every National Health Service (NHS) trust that will recruit patients. With individualised applications and separate review processes, the local NHS trust approvals are increasing the cost and complexity of initiating clinical trials.[3] A parliamentary report on clinical trials in September 2013 highlighted separate local NHS R&D approvals as the biggest barrier to initiating clinical trials in the UK and one of the factors that led to a 22% decline in the number of clinical trials conducted in the UK between 2007 and 2011.[4]

To obtain local R&D approval a named principal investigator and site need to be submitted to ethics through the Integrated

## BACKGROUND

Many researchers have communicated frustration at the length of time taken to open

Research Application System (IRAS) either as part of the original trial application or as a substantial amendment afterwards. While processing it as a substantial amendment can be a little onerous, approval letters are usually issued within a matter of days.

Following this a Site Specific Information (SSI) Form needs to be completed within IRAS summarising the trial, what local resources are needed and who will be involved. Supporting documents are attached such as Patient Information Sheets (PIS) and Consent Forms containing the trust logo, and CVs of all site staff performing trial duties. Once the principal investigator has signed the completed SSI it is electronically submitted to the relevant R&D department in England or manually emailed to trusts in constituent nations who do not use the IRAS system.

The time taken for local R&D departments to approve the application has been the primary focus of criticism.[1][3][5–7] In 2011 Whitehead et al[7] reported that approval took between 6 and 197 days (mean 42 days) from SSI submission. In 2007 Al-Shahi Salman[6] retrospectively interviewed R&D departments from four trials and identified a median delay of 44 working days (IQR 23.0–80.0). With inconsistent approaches across trusts, duplication of ethical review and unclear delays the research community called for agreed timeframes to be imposed on R&D departments.[1][6][8]

In 2011 the government introduced NHS contract benchmarks to incentivise trusts to recruit the first patient within 70 days of receiving a valid SSI application.[9] Alongside this, internal targets from the NIHR Clinical Research Network (NIHR CRN) required English trusts to approve valid applications within 30 days. These internal NIHR CRN targets contributed to the public High Level Objectives that include a median approval time across all trial sites of 40 days.

This study seeks to prospectively assess the time taken to open UK research sites now these targets are in place and identify reasons for any remaining delays.

## METHODS
SANADII (http://www.sanad2.org.uk) is a phase IV pragmatic study comparing standard and new treatments for patients with epilepsy, which aims to recruit 1510 participants over the age of 5 from over 100 sites across all four constituent nations of the UK.

The trial follows routine care for newly diagnosed epilepsy patients, with the exception that patients complete a postal questionnaire at 3 and 6 months in the first year, and annually thereafter. Patients are followed up for two to five and a half years depending on when they are recruited into the trial.

The additional cost of patients receiving the new drugs compared with the standard drugs (the Excess Treatment Costs), have been calculated at an average total cost of £1182 per patient.

SANADII is funded by the NIHR Health Technology Assessment Programme (HTA), cosponsored by the University of Liverpool and The Walton Centre and is managed by the Clinical Trials Research Centre (CTRC) within the University of Liverpool. SANADII funding was confirmed on the 17 May 2011 (subject to contract). The ethics application was submitted on 18 April 2012 and received approval on 7 June 2012.

During data collection the study had a full time trial coordinator from March 2012 and up to three trial coordinator assistants, all of whom were involved in opening research sites. Alongside this the NIHR Medicine for Children Research Network, the NIHR Dementias and Neurodegeneration Research Network and the Welsh Epilepsy Research Network allocated staff time to actively help set up sites identified through their contacts.

### Inclusion criteria
All SANADII research sites listed on the original ethics trial application or any subsequent additional sites were included in the data collection. This included 17 original SANADII sites that received ethical approval in June 2012, and another 111 sites added in 2013 across four ethics amendments (3 January 2013, 3 April 2013, 15 July 2013 and 26 November 2013).

A number of NHS trusts who planned to recruit both children and adults requested that paediatric and adult research teams apply for separate R&D approval as there was little or no interaction between them due to geography or clinic schedules. Where teams completed separate SSI forms for the same trust, they have been analysed as separate sites.

### Data collection
SANADII trial management staff were trained by the authors to record key milestone dates in a purpose built web database. Data were collected from the date a site received ethics approval until they no longer wished to participate or they opened to recruitment (figure 1). Alongside milestone dates trial staff also recorded delay reasons using a coding system created to cover a variety of potential issues across all stages of the site set up process (see online supplementary table S1). CTRC staff were required to enter a minimum of one delay reason a fortnight, but could enter up to three delays at any given point. They were encouraged to enter the same codes until the delay was resolved or other delays were perceived to have a greater impact on trial site progress.

Key site characteristics were logged such as the constituent nation, whether they were recruiting adults, children or both and whether a research network was actively working with the SANADII team to get all the paperwork and local approvals completed.

Where research networks were working directly with sites, trial staff would complete data entry based on fortnightly updates provided by the network staff.

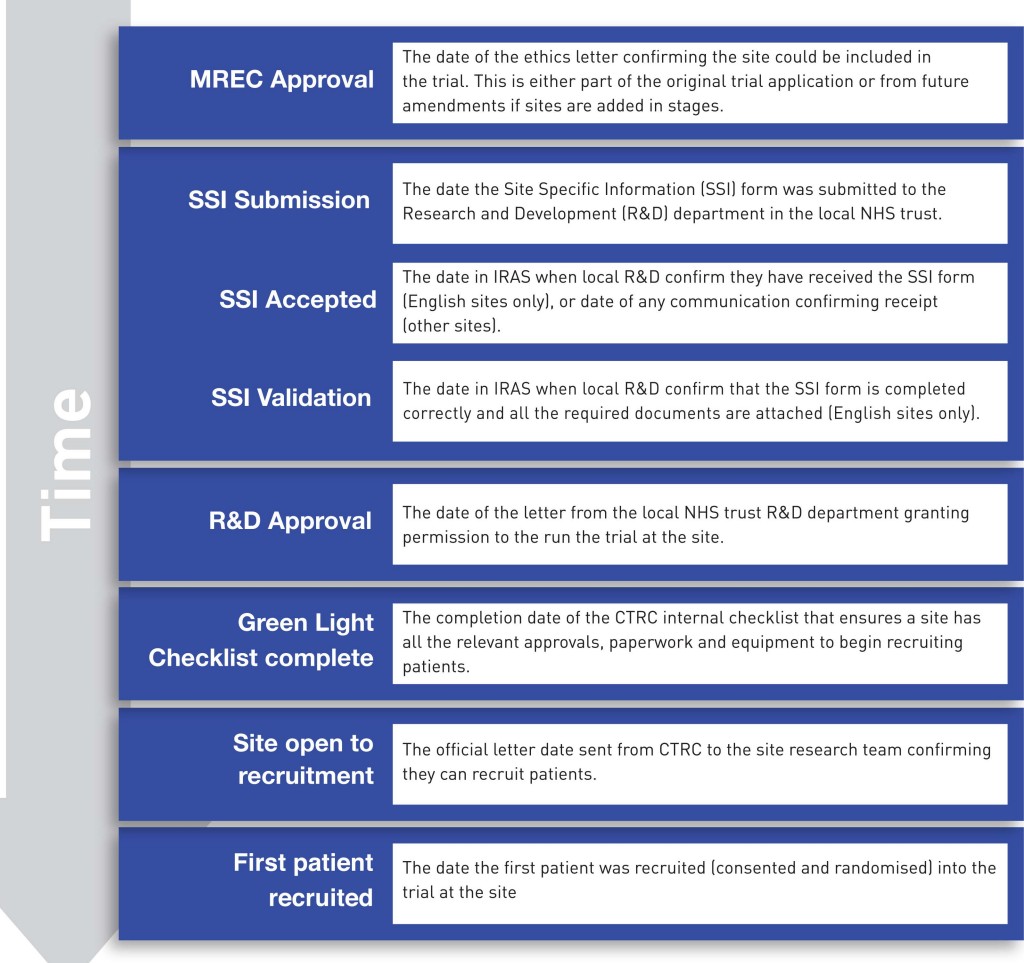

**Figure 1** Milestones and definitions. Definitions of key milestone dates collected during the site set up process.

At the point of analysis SANADII trial management staff also provided separate data on the date the first patient was recruited at each site.

Milestone data for other CTRC trials initiated in 2012 and 2013 were retrospectively collected from trial records for comparison with SANADII data.

### Identifying delay reasons

Where sites took over a specified time to reach milestones (>3 months from MREC to SSI submission, >30 days from SSI validation to R&D approval, or >30 days from R&D approval to opening to recruitment) the longest continuous delay reason recorded between those two milestone dates was identified.

The original data were analysed to identify the top three longest delays for each site milestone and then manually reviewed by one author (AK). If necessary, emails, trial records and discussions with SANADII staff were used to confirm or revise the longest delay code.

As a result of SANADII staff working on other site set up activities while R&D reviewed the SSI application, delay reasons between the milestones SSI validation and R&D approval were predominantly identified from trial records and emails retrospectively.

### Statistical analysis

Milestone validation checks and all analyses were performed using SAS V.9.3. All of the outcomes were measured using a time-to-event survival analysis approach. Sites that had not yet had observed outcomes were censored at the date of data extraction. The outcome data were compared across factors (recruitment of children and/or adults, hospital type, network support compared with no network support) using Kaplan-Meier curves and the p values from log-rank tests with relative effects of factors summarised using median times with 95% CIs obtained from the Kaplan-Meier plots, and HRs with 95% CIs. In addition, 25% and 75% quartiles with 95% CIs obtained from the Kaplan-Meier analyses are presented. The study did not require ethical approval.

### RESULTS

The first trial sites received MREC approval on the 7 June 2012 and data were extracted from the database on

the 7 March 2014. SANADII had 128 sites with ethical approval of which 32 (table 1) had dropped out before submitting their SSI form (18% of adult sites and 28% of paediatric teams) leaving 96 participating sites (see online supplementary table S2).

Overall, participating sites demonstrated a median opening time of 10.5 months (IQR 7.3–15.2) from MREC approval. However, the first 17 SANADII sites, who received ethical approval in June 2012, had a median opening time of 14.5 months (IQR 11.4–16.0; figure 2) as they experienced a unique 6-month delay as excess treatment costs needed to be approved by hospital trusts and primary care. With the NHS restructure and the formation of Clinical Commissioning Groups (CCG) only coming into effect in April 2013 no-one was willing to review applications, putting all 17 sites on hold until solutions began to emerge in January 2013. Fifteen of the first 17 sites are participating sites and consequently we have excluded these 15 sites from the analysis of the time from MREC approval to site opening. For fourteen of these sites the delay occurred pre-SSI submission and so these 14 have also been excluded from

analysis of time from MREC approval to SSI submission. For one site the delay was post R&D approval and so it has been excluded from analysis of time from R&D approval to site opening and from SSI validation to recruiting the first patient

The remaining 81 participating sites who received their ethical approval throughout 2013 had a median opening time of 9.7 months (IQR 6.2 to Not Reached) (figure 2).

Breaking down the site set up process we observed that the median time from MREC approval to SSI submission was 7 months (IQR 4.1–12.3), from SSI Validation to R&D Approval 16 days (IQR 5.0, 32.0) and from R&D Approval to opening 15 days (IQR 8.5–40.0; figure 3). Thirty three of 45 sites (73%) received R&D approval within 30 days of the SSI being validated (see online supplementary table S5).

Analysis of site characteristics showed there was no difference in site opening times between teaching hospitals and district general hospitals with medians 9.7 vs 10.0 months, HR 0.99 95% CI (0.5 to 2.0) p=0.98 (see online supplementary figure S1). However, at any point

**Table 1** SANADII site characteristics at 7 March 2014

| | |
|---|---|
| **Total sites with MREC approval:** | **128 (%)** |
| Status: | |
| Participating | 96 (75) |
| No longer participating | 32 (25) |
| Participating status: | |
| (1) Had MREC approval but not yet submitted SSI | 31 (32) |
| (2) Submitted SSI but not yet had R&D approval | 8 (8) |
| (3) R&D approved but Green Light Checklist not yet completed | 4 (4) |
| (4) Green Light Checklist completed but not yet open to recruitment | 0 (0) |
| (5) Site open but not yet recruited first patient | 15 (16) |
| (6) Site open and first patient recruited | 38 (40) |
| Recruitment type (of those participating) | |
| Adult | 47 (49) |
| Paediatrics | 43 (45) |
| Both paediatrics and adult | 6 (6) |
| Hospital type (of those participating) | |
| District general hospital | 56 (58) |
| Health centre | 2 (2) |
| Specialist paediatrics | 6 (6) |
| Specialist tertiary | 1 (1) |
| Teaching hospital | 31 (32) |
| Country (of those participating) | |
| England | 76 (79) |
| Isle of man | 1 (1) |
| Northern Ireland | 2 (2) |
| Scotland | 7 (7) |
| Wales | 10 (10) |
| Set up actively supported by Research Network (of those participating): | |
| NIHR Comprehensive Local Research Network (CLRN)* | 4 (4) |
| NIHR Dementias and Neurodegeneration Network (DenDRoN)* | 14 (14) |
| NIHR Medicines for Children Research Network (MCRN)* | 24 (25) |
| Welsh Epilepsy Research (WERN) | 6 (6) |
| No Network | 48 (50) |

*Networks within NIHR Clinical Research Network before it was restructured in April 2014.
MREC, Multicentre Regional Ethics Committee; SSI, Site Specific Information; R&D, Research and Development.

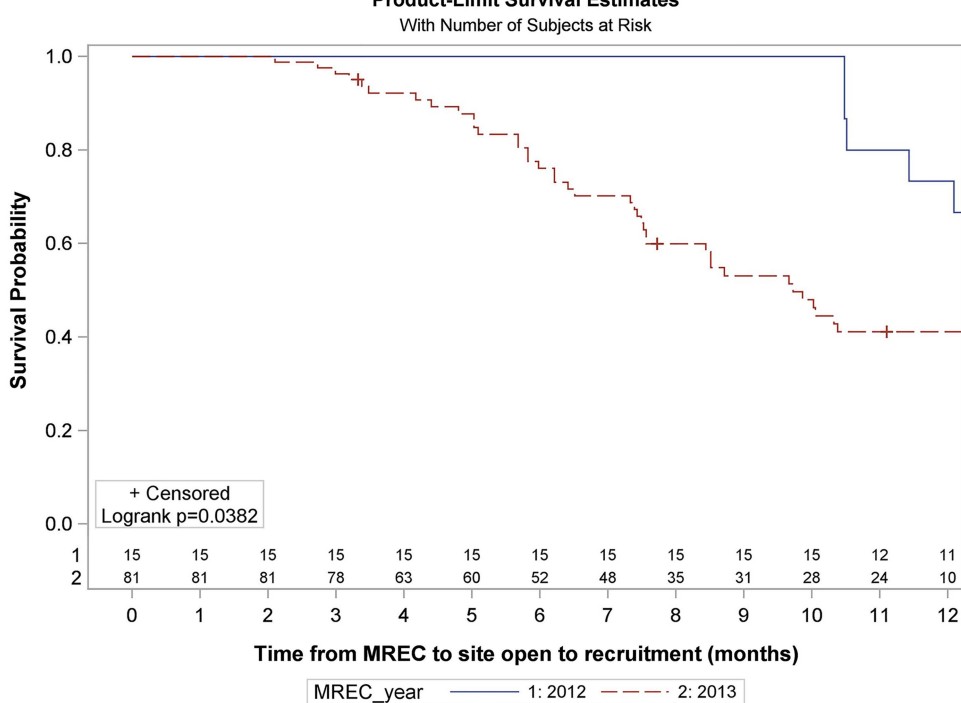

**Product-Limit Survival Estimates**
With Number of Subjects at Risk

**Figure 2** Survival analysis of time from MREC approval to site open to recruitment for participating sites (months). Numbers at risk are displayed above the x axis. Two of the 17 sites which received ethical approval in 2012 and 30 of the 111 sites who received ethical approval in 2013 have been excluded from the analysis as they are no longer participating. NR= not reached. For the full Kaplan-Meier summary see online supplementary table S3.

paediatric sites were opening twice as quickly as adult sites with medians; 8.5 months vs Not Reached; HR 2.33 (95% CI 1.1 to 4.8); p=0.02 (see online supplementary figure S1). This may have been influenced by the strong backing of the NIHR Medicine for Children's Research Network (MCRN) who coadopted the trial. The NIHR MCRN network was one of the three networks who dedicated staff time to help open sites and engaged their Local Research Networks to help nationally. Twenty-five paediatric sites (60% of all paediatric sites) compared with 14 adult sites (40% of all adult sites) had active network support.

Of the 69 sites that took over 3 months to submit their SSI form the most frequent continuous delays were negotiating excess treatment costs (n=16), finalising logistics at site (n=11), collecting research Curriculum Vitae (CVs) (n=10) and ongoing participation discussion (n=10). During data collection SANADII staff were aware that some R&D departments were holding off SSI submission in order to review trials before they were formally submitted and to meet their targets. R&D departments holding off submission accounted for the longest delay in five sites (table 2).

Of the 12 sites that took over 30 days to be granted R&D approval, waiting for the contract to be signed by site (n=3) and unexplained delays (n=3) were the most frequent reasons (table 2).

Eighteen sites took longer than 30 days to open to recruitment from R&D approval with most frequent reasons including getting the delegation log signed

(n=5), waiting for training on trial procedures (n= 4), and waiting for the contract to be signed by site (n=2). In another two sites the R&D approval letter itself held up site opening when it was sent to trial staff a month after it was issued (table 2).

The median time for recruiting the first patient from SSI validation was 127 days (IQR 64.0–230.0) which is significantly more than the 70 day contract benchmark (see online supplementary figure S2). Only 12 of 40 sites (30%) that have recruited their first patients did so within 70 days of SSI validation (see online supplementary table S5). Delay reasons were not formally collected after the site was open to recruitment. However, as part of the trial management process sites were contacted to see if there were any problems with recruitment. In general sites were not able to give a definitive reason why there was a delay but responses included a lack of eligible patients, holidays or that there was no problem and they would recruit patients shortly. However, new epilepsy is a common presentation and trial follow-up was assessed through routine care, so recruiting patients should be relatively quick and easy in comparison to many other trials.

Milestone data were retrospectively collected for another four CTRC studies and compared with SANADII demonstrating that in terms of the time to open sites, these findings are not unique (see online supplementary figure S3). The cohort of studies includes a medical device trial (BASICS), another drug trial (TAILOR), a microbiology study (DINOSAUR Micro.) and a service

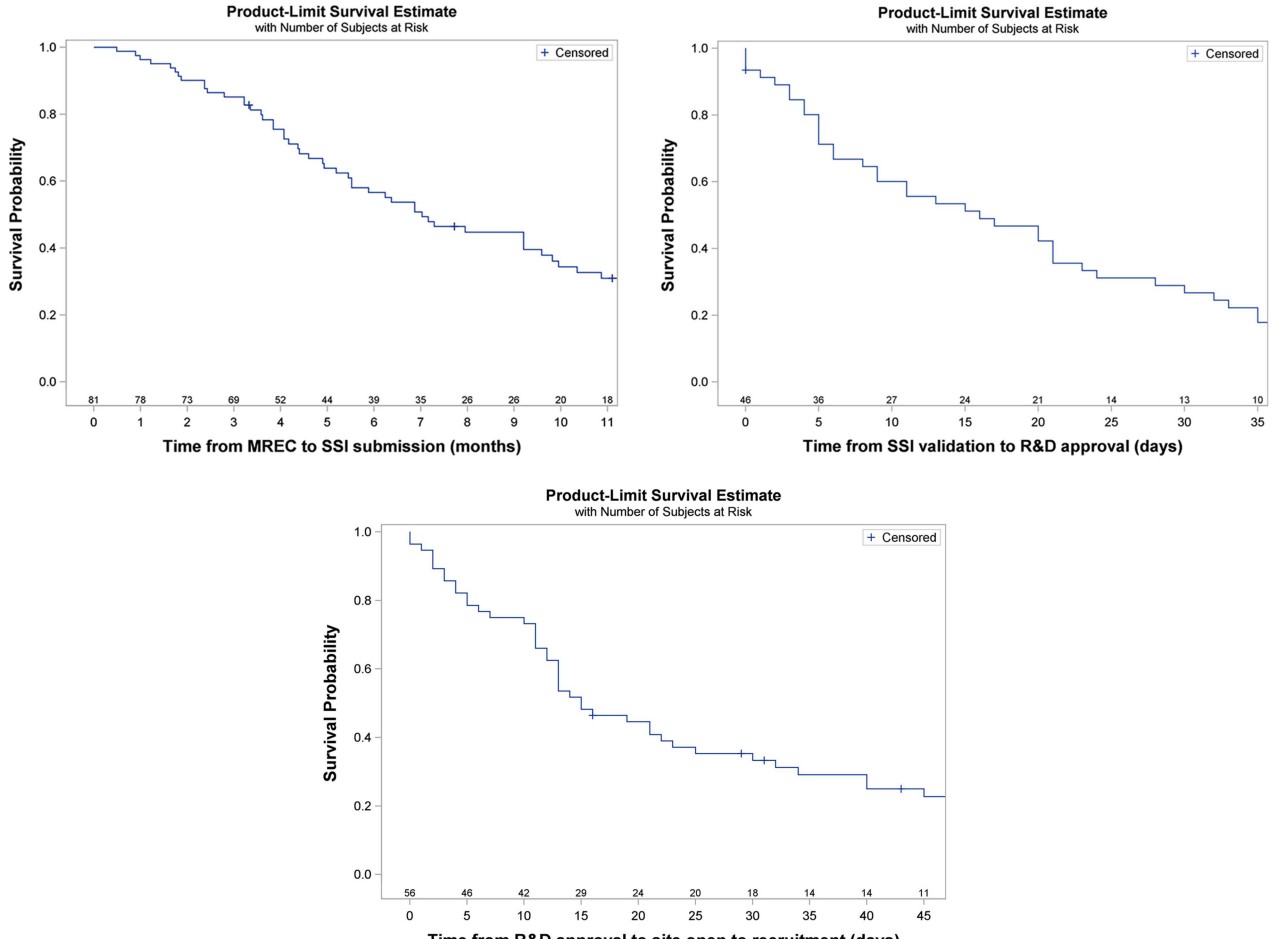

**Figure 3** Survival analysis of time between milestones for participating sites. Numbers at risk are displayed above the x axis. *Time from MREC to SSI Submission*: 15 of the 96 participating sites have been excluded from the analysis because 14 of them experienced unique delays due to the NHS restructure and 1 site has a missing SSI submission date. NR= not reached. *Time from SSI validation to R&D Approval*: 50 of the 96 participating sites have been excluded from the analysis. Thirty-one sites have not submitted the SSI form, 7 have submitted SSI form and not received R&D approval yet but have a missing SSI validation date, 11 have received R&D approval but have a missing SSI validation date and 1 site has a negative time due to technical issues with IRAS. *Time from R&D Approval to Site opening*: 40 of the 96 participating sites have been excluded because 39 sites have not reached R&D approval and 1 site with ethical approval in 2012 was excluded as they experienced unique delays due to the NHS restructure. For the full Kaplan-Meier summary tables see online supplementary table S4.

evaluation (DINOSAUR SE), for which site set up started in 2012 or 2013. The lowest median opening time was 5.3 months (IQR 4.8–7.4) which was for the service evaluation which did not require local R&D approval.

## DISCUSSION
### Principal findings
Government initiated targets and NIHR CRN initiatives are impacting the time taken to gain local R&D approval from submission of valid documentation. Where previously researchers were left waiting an indeterminate amount of time, the median approval has been reduced from around 40–45 days to 16 days with 73% of applications being approved within 30 days. Researchers now have an understanding of when they are likely to receive approval and so can proactively align other activities to ensure that sites open to recruitment shortly afterwards.

However, it is concerning that despite R&D departments approving applications more quickly, sites still took more than 9 months to open with the majority of time spent preparing to submit the application form. The most frequent reason identified before SSI submission was negotiating excess treatment costs which was the longest delay in 16 (23%) sites but also contributed to delays in 39 (48%) sites with MREC dates in 2013.

Excess treatment costs are the difference in cost between the investigative treatments in a trial and standard care. The NHS is required to cover normal treatment costs, but additional costs incurred by trials (including service support costs and research costs) need to be correctly attributed to either research funders or specific trust or CCG budgets. Previous authors have highlighted attributing and negotiating costs as a cause for delays in setting up research sites.[3 6 7]

**Table 2** Frequency of longest continuous delay reasons for SANAD-II sites that took over a specified time to complete milestones

| | |
|---|---|
| **MREC to SSI Submission >3 months** | |
| Excess Treatment Cost Negotiation | 16 |
| Finalising logistics *ie, SSI completion, identifying staff*\* | 11 |
| Collecting CVs | 10 |
| Participation Discussions: *ie change of PI, ongoing feasibility*† | 10 |
| Not a priority site for SANADII staff‡ | 7 |
| Awaiting R&D permission to submit SSI§ | 5 |
| Waiting for PI to sign SSI | 3 |
| Resolving R&D query/s | 2 |
| Pharmacy approval | 1 |
| Local ethics | 1 |
| Site research team holding off set up¶ | 1 |
| No news from network | 1 |
| No response to email\*\* | 1 |
| | 69 |
| | |
| **SSI Validation to R&D Approval >30 days** | |
| Waiting for contract to be signed by site | 3 |
| Unknown delay | 3 |
| Waiting for training on trial procedures | 1 |
| Contract being signed by sponsor | 1 |
| Contract review by site | 1 |
| Excess Treatment Cost Negotiation | 1 |
| Governance query | 1 |
| Waiting for staff to come into post | 1 |
| | 12 |
| | |
| **R&D Approval to Opening > 30 days** | |
| CTU waiting for completed delegation log | 5 |
| Waiting for training on trial procedures | 4 |
| Waiting for site to sign contract | 2 |
| Waiting for R&D approval letter | 2 |
| CTU waiting for Acknowledgement receipt of Essential Documents | 1 |
| Waiting for sponsor to sign contract | 1 |
| Waiting for staff to come into post | 1 |
| Waiting for staff to do GCP training | 1 |
| No response to email§ | 1 |
| | 18 |

Where some delay reasons have been grouped the text in italics indicates the most frequent codes used in the grouping. The wording of some delay code names for smaller frequencies have been adapted to give better understanding of the actual delays based on notes made at data collection.
\*Included SSI completion (n=9), identifying staff (n=1) and agreeing recruitment target (n=1).
†Included change of PI (n=8), feasibility discussions (n=1), decided not to participate but are now participating again (n=1).
‡This reason increased from interim analysis in October due to the reduction in staff (trial coordinator assistants) towards the end of 2013. As a result trial staff prioritised proactive sites and weren't able to follow-up less proactive or new sites.
§R&D holding off submission possibly due to internal issues (n=1), R&D were reviewing the SSI before formal submission (n=3) and research network informed us they were waiting for R&D permission to submit the SSI (n=1).
¶Research staff at site did not have capacity and requested that contact be re-established in 3 months.
\*\*No response from clinical staff.
CVs, Curriculum Vitaes; SSI, Site Specific Information; R&D, Research and Development.

On several occasions trial staff were asked to help calculate costs, prepare funding applications and give advice on how to resolve excess treatment costs. While the Department of Health's AcoRD guidance[10] is useful in defining the different types of research costs and who is broadly responsible for them it seems to have done little to ease the problem in practice. Dr Greta Westwood recently wrote in the Health Services Journal, 'Successive policy documents remind NHS organisations of the duty to fund excess treatment costs, yet no national scheme exists to manage them'.[11]

Finalising site logistics and ongoing participation discussion together accounted for the longest delay in 21 (30%) sites. Both of these highlight the challenges of working with busy healthcare professionals who have competing priorities. It has been suggested that the level of data collection required under the current governance process is unmanageable for clinicians and should be revised to prevent a decline in participation.[12] Finalising site logistics mainly compromised of delay codes about SSI completion. However, in SANADII the majority of the SSI form was prepopulated before sending to clinical staff

for review but could take up to several weeks or even months to complete. This perhaps reflects the time needed to discuss the trial with managers and affected departments (eg, pharmacy), agree staffing and confirm capacity, all of which needs to be documented in the SSI form. Ongoing participation discussion included delay codes for feasibility assessments post-ethical approval and changes of Principal Investigators. We found a number of Principal Investigators decided not to participate requiring us to identify replacements and gain new ethics approval before submitting the SSI, all adding to the time needed to open the site.

The third most significant delay reason was collecting research CVs. IRAS guidance requires that short research CVs are attached to the application for anyone who will perform trial related duties. However, our experience shows it can take months to collect these from research staff. According to Good Clinical Practice guidelines, CVs of investigators and subinvestigators should be collected and filed by both the site and sponsor before the trial commences to 'document qualifications and eligibility to conduct a trial and/or provide medical supervision of participants'.[13] Our concern is that sponsors or clinical trial units (under delegated authority from the sponsor) are unable to and do not formally assess this despite collecting CVs. We believe consideration should be given to a process in which the employing trust is responsible for confirming that site staff are suitably qualified and ensure they hold adequate documentation to demonstrate this.

In the case of HTA funded trials such as SANADII, significant resource is spent on staff in order to negotiate NHS systems, which is very difficult to justify in a system with limited resources that must be spent wisely. It is worth noting that the CTRC invested a larger number of staff hours than had been accounted for in the grant application to help facilitate site opening. From January to October 2013 SANADII had a full time trial coordinator and the equivalent of three full time trial coordinator assistants whose main priority was to open sites. This was reduced to one coordinator and the equivalent of one full time assistant towards the end of 2013 due to funding constraints. SANADII also received strong support from a number of topic specific research networks, three of whom dedicated staff time to opening sites alongside the SANADII staff.

Despite this investment it still took on average 9.7 months to open sites and the official start of the trial was delayed by 4 months. While the trial is currently exceeding its original recruitment schedule and is not likely to need a time extension, many other studies have[14 15] and the costs of managing delays are being passed onto funders. Future grant applications will need to build in increased staffing costs if the governance process is not successfully streamlined.

## Meaning of the study: possible explanations and implications for clinicians and policymakers

On the 31 March 2014, the Health Research Authority (HRA) announced that they had approval to continue streamlining the governance process and implement a single application system[16] as recommended in the Academy of Medical Science Review.[17] The HRA's summary plan support the results of this study stating 'Although these targets and benchmarks have been successful in driving reductions in timelines, the metrics for REC and R&D systems are in isolation, and analysis now shows that without integration there is risk that any further downward pressure on timelines will simply move delays around the system'.[18]

The HRA plans to incorporate ethical review and the legal aspects of the current R&D approval process at sites into one global permission. They believe this will remove a lot of repetition at a local level and instead will give trusts more time to work on setting up of the study.[18]

This is a welcome development but to successfully reduce clinical governance times the authors encourage review of the issues raised in this report. It is essential that any change of governance not only revise how approval is obtained, but also holistically consider what documents and processes are needed. Under the new streamlined process local site permission will still need to be granted based on capacity and research CVs will still need to be collected prior to study start under Good Clinical Practice (GCP) and International Conference of Harmonisation (ICH) guidelines. A change to trusts centrally holding and disseminating up-to-date research CVs may be one pragmatic solution.

## Strengths and weaknesses of the study

Milestones were clearly defined (figure 1) at the start and data was manually reviewed against stored documents with validation checks in SAS used to minimise data entry errors. Consequently the milestone data provides a reliable, objective basis for analysing time taken to open research sites.

The study is limited in that it recorded the CTRC's perspective of delays. We have not requested reasons from clinical staff or R&D departments other than would normally occur in day to day trial management. Similarly the process of choosing delay reasons was subjective although trial management staff were trained to enter data consistently.

Delay reasons for SSI validation to R&D approval should be interpreted with some caution due to the lack of communication from R&D departments about progress when reviewing the application.

## Unanswered questions and future research

If HRA performance targets become based on time from ethical submission to opening sites there is a potential danger that delays may simply be shifted to an earlier part of the process as seen with the introduction of the NHS contract benchmarks. Delaying ethics submission in order to meet targets would be particularly challenging for researchers as many funders only begin releasing grant awards on receipt of ethical approval.

Future research will need to review site opening times and delays from grant awards onwards in order to assess the effect of the HRA's new streamlined approval process.

This case study provides a good baseline to evaluate the true impact of future changes by the HRA. It would be beneficial to repeat the study in a few years to compare results and assess whether timeframes have increased or diminished.

## CONCLUSION

Reducing the length of time needed to set up clinical research sites in multicentre studies continues to be a high priority in order for the UK to remain a competitive market for research.

The HRA's proposed streamlining into a single application is a clear step forward, but is awaited with a sense of caution to see what difference it will make given the nature of the delays identified in this study.

Alongside governance changes a number of issues need to be addressed including implementing a clear strategy to manage excess treatment costs in an increasingly budget constricted NHS. The collection of research CVs needs reviewing in order to truly achieve the desired effect of protecting patients without burdening busy clinicians with unproductive paperwork. Finally additional resources need to be invested into helping clinical staff and hospital management quickly and decisively incorporate trials into their ongoing workflow.

This study highlights the current variety of delays that affect trial site set up and the impact that these have on the use of resources and grant funding. To make significant reductions in site opening times will require understanding, close co-operation and investment from a wide range of stakeholders including NHS Trust and CCG governance staff, clinical staff, the HRA and the newly restructured NIHR Clinical Research Networks all of whom can influence and support different aspects of the process.

**Contributors** AK developed the study protocol and database specification, trained trial managers on data entry, manually reviewed data during the analysis and drafted the manuscript and she was part of the SANADII trial management staff between January and October 2013. PW conceived the research, provided feedback on the specification, led on the analysis plan and commented on the manuscript. CG provided feedback on the protocol and commented on the manuscript. AGM is Chief Investigator of SANADII. He provided feedback on the protocol and commented on the manuscript. HH provided feedback on the protocol and commented on the manuscript. AM developed and performed the validation checks and statistical analysis in SAS and commented on the manuscript. SB is the SANADII trial Co-ordinator and was involved in facilitating data collection and answering trial related queries raised during the data analysis as well as commenting on the manuscript. All the authors critically revised the draft for important intellectual content, and gave final approval of the version to be published. PW is the guarantor for the study and affirms that this manuscript is an honest, accurate and transparent account of the study being reported; that no important aspects of this study have been omitted; and that any discrepancies from the study as planned (and, if relevant, registered) have been explained.

**Funding** This study was funded by the MRC North West Hub for Trials Methodology (grant number G0800792). SANADII is funded by the NIHR Health Technology Assessment Programme (grant number 09/144/09).

**Competing interests** All authors have completed the ICMJE uniform disclosure form at http://www.icmje.org/coi_disclosure.pdf and declare: no support from any organisation for the submitted work; AGM received research funding from Consortium of UCB Pharma, GSK, Eiasi and Viropharma for a seizure management audit and presented a lecture on epilepsy treatment for Sanofi Aventis; no other relationships or activities that could appear to have influenced the submitted work.

**Provenance and peer review** Not commissioned; externally peer reviewed.

**Data sharing statement** No additional data are available.

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
