## [Reviewer comments · BMJ Open]

Some articles will have been accepted based in part or entirely on reviews undertaken for other BMJ Group journals. These will be reproduced where possible.

ARTICLE DETAILS

TITLE (PROVISIONAL)	Opening research sites in multicentre clinical trials within the UK: A detailed analysis of delays.
AUTHORS	Kearney, Anna; McKay, Andrew; Hickey, Helen; Balabanova, Silviya; Marson, Anthony; Gamble, Carrol; Williamson, Paula

VERSION 1 - REVIEW

REVIEWER	Peter sandercock University of Edinburgh
REVIEW RETURNED	30-Jun-2014

GENERAL COMMENTS	a very valuable paper provided up-to-date on the current state of clinical trial bureaucracy in the UK. The paper provides some reassurance that things have improved, but there is still a long way to go. The delays described in this paper are still leading to a shameful waste of precious NHS research resource, and even more important, loss of enthusiasm for research by site PI's.
--

REVIEWER	Nancy Lester Business Development Director National Institute for Health Research Clinical Research Network (NIHR CRN) Leeds, England
REVIEW RETURNED	10-Jul-2014

GENERAL COMMENTS	Specific comments on review section Section 4 - Note: the research might need to be repeated in a few years' time to see if delays have increased or diminished. (using same methodology but a different but similar type of study) Section 7 - Please note I am not a methodological / statistics expert. I have completed section 7 based on my current (limited) knowledge. Section 13 - I am not clear about this section so have left it blank. Section 15 - A few typos / minor inconsistencies of punctuation etc, e.g. p.3:18 cf p.3:21 and p.4:52 cf p.4:54; pg 9:5 these 14 'have' not 'of.' General Comments on Document and/or Areas for Clarification
--

Whilst a number of the issues highlighted in this paper are frequently cited by researchers as reasons for delays to the delivery of clinical trials, from my experience much of this is based on anecdote. There is little specific formal evidence and robust data collected in a systematic and methodical robust way to support this. This case study helps address this.

Reference to research and development approvals is made throughout the document. It would be helpful to define what is meant by local NHS Research and Development approval. The House of Commons Science and Technology Committee: Clinical Trials report (ref 3) refers to this as the requirement for researchers to obtain separate approvals for each NHS organisation involved.

The reference to networks (Background Section) should be updated to reflect the new NIHR Clinical Research Network Structures introduced on 1 April 2014.

Also, I suggest that the various references to the involvement of the NIHR Clinical Research Network (NIHR CRN) throughout the document are reviewed to ensure greater clarity and consistency (page 5;6-7; page 6;48-49; page 13;12-14). The NIHR Medicines for Children Clinical Research Network and the NIHR Comprehensive Clinical Research Network are both part of the NIHR CRN, for example.

It would be helpful to clarify if the network support referred to on page 13:12-19 is the same or in addition to that referred to earlier in the manuscript (methods section page5:2-7)? This should be clearer.

There is evidence to show that real and sustained improvements are being made in some areas cited as delays. For example the NIHR CRN has achieved a range of improvements to support more effective study delivery. This year (13/14) the CRN has achieved a 43% improvement in the time taken for study set-up for the studies it supports, with the median number of days to achieve NHS Permission for all study sites 25 calendar days. Further details can be found at the following link: <http://www.crn.nihr.ac.uk/annualstats>

Discussion of principal findings (pg16) / Abstract conclusion (pg2) – Other initiatives have been introduced and are becoming embedded across NHS. It could be argued that these broader initiatives, not just targets, support improvements in study set up. For example NIHR CRN – see above.

Reference to list of where additional costs incurred by trials should be attributed to (pg16; 30-37) should include Research Funders

Whilst this report considers all UK countries, reference to NIHR CRN relates to the NHS in England only.

Reference to Health Technology Assessment programme (pg4:49) should be NIHR Health Technology Assessment Programme.

VERSION 1 – AUTHOR RESPONSE

Reviewer Comments:

Section 4 - Note: the research might need to be repeated in a few years' time to see if delays have increased or diminished. (using same methodology but a different but similar type of study)

- We have reworded our final paragraph in the 'unanswered questions and future research' section to affirm that it would be beneficial to repeat the study in order to assess the HRA's streamlining.

Section 15 - A few typos / minor inconsistencies of punctuation etc, e.g. p.3:18 cf p.3:21 and p.4:52 cf p.4:54; pg 9:5 these 14 'have' not 'of.'

- The typos and inconsistencies outlined in section 15 have been corrected

Areas for Clarification:

1. Reference to research and development approvals is made throughout the document. It would be helpful to define what is meant by local NHS Research and Development approval. The House of Commons Science and Technology Committee: Clinical Trials report (ref 3) refers to this as the requirement for researchers to obtain separate approvals for each NHS organisation involved.

- To help clarify, we have switched the second and third sentences within the background section so that local research and development approval is described at the start of the publication. (Please note the track changes were accepted to enable the Endnote references to update, so I have highlighted the relevant paragraphs instead.) The description has also been developed to be more consistent with the House of Commons Science and Technology Committee report, emphasising these are separate approvals from every NHS Trust that will recruit patients.

2. The reference to networks (Background Section) should be updated to reflect the new NIHR Clinical Research Network Structures introduced on 1 April 2014.

- The background section has been updated to communicate that the internal targets are from the NIHR CRN and we have removed reference to the old Comprehensive Local Research network.

3. Also, I suggest that the various references to the involvement of the NIHR Clinical Research Network (NIHR CRN) throughout the document are reviewed to ensure greater clarity and consistency (page 5;6-7; page 6;48-49; page 13;12-14). The NIHR Medicines for Children Clinical Research Network and the NIHR Comprehensive Clinical Research Network are both part of the NIHR CRN, for example.

- We have added 'NIHR' to relevant network references to distinguish between those who are part the NIHR CRN network and those that are not. In Table 1 we have also given the networks their full title and added a footnote to clarify that these were research networks within the NIHR CRN before it was restructured in April 2014.

4. It would be helpful to clarify if the network support referred to on page 13:12-19 is the same or in addition to that referred to earlier in the manuscript (methods section page5:2-7)? This should be

clearer.

- On page 5 we have listed the networks that dedicated staff time to opening up research sites, and on page 13 we have clarified that the NIHR MCRN was one of the three listed earlier in the manuscript.

5. There is evidence to show that real and sustained improvements are being made in some areas cited as delays. For example the NIHR CRN has achieved a range of improvements to support more effective study delivery. This year (13/14) the CRN has achieved a 43% improvement in the time taken for study set-up for the studies it supports, with the median number of days to achieve NHS Permission for all study sites 25 calendar days. Further details can be found at the following link: <http://www.crn.nihr.ac.uk/annualstats>

Discussion of principal findings (pg16) / Abstract conclusion (pg2) – Other initiatives have been introduced and are becoming embedded across NHS. It could be argued that these broader initiatives, not just targets, support improvements in study set up. For example NIHR CRN – see above.

- We have updated the principal findings to credit NIHR initiatives for their involvement in reducing the time taken for local R& D to approval valid applications. Throughout the publication we feel we have made repeated reference to the positive involvement of the NIHR in facilitating site set up. The strong engagement of the NIHR MCRN networks is believed to have influenced the speed of paediatric site set up (Page 13), and the study support offered by NIHR DenDRoN and the old NIHR CLRN's is clearly documented throughout the article.

6. Reference to list of where additional costs incurred by trials should be attributed to (pg16; 30-37) should include Research Funders

- 'Sponsors' has been changed to 'research funders' in line with the reviewer's comments and to reflect that the primary role of sponsors is trial oversight not funding.

7. Whilst this report considers all UK countries, reference to NIHR CRN relates to the NHS in England only.

- The background section has been updated to show that the NIHR CRN targets are aimed at English sites.

8. Reference to Health Technology Assessment programme (pg4:49) should be NIHR Health Technology Assessment Programme.

- Correction has been made in line with the reviewer's comments